# Encapsulation of Pineapple Peel Extracts by Ionotropic Gelation Using Corn Starch, *Weissella confusa* Exopolysaccharide, and Sodium Alginate as Wall Materials

**DOI:** 10.3390/foods12152943

**Published:** 2023-08-03

**Authors:** Anna María Polanía, Cristina Ramírez, Liliana Londoño, German Bolívar, Cristobal Noe Aguilar

**Affiliations:** 1MIBIA Group, Biology Department, Faculty of Natural and Exact Sciences, Universidad del Valle, Cali 760031, Colombia; anna.polania@correounivalle.edu.co (A.M.P.); cristina.ramirez@correounivalle.edu.co (C.R.); german.bolivar@correounivalle.edu.co (G.B.); 2Food Research Department, School of Chemistry, Universidad Autónoma de Coahuila, Saltillo 25280, Coahuila, Mexico; 3BIOTICS Group, School of Basic Sciences, Technology and Engineering, Universidad Nacional Abierta y a Distancia—UNAD, Palmira 763531, Colombia; liliana.londono@unad.edu.co

**Keywords:** ultrasound-assisted extraction, phenolic compounds, pineapple peel, encapsulation, ionotropic gelation

## Abstract

Phenolic compounds that are present in pineapple by-products offer many health benefits to the consumer; however, they are unstable to many environmental factors. For this reason, encapsulation is ideal for preserving their beneficial effects. In this work, extracts were obtained by the combined method of solid-state fermentation with *Rhizopus oryzae* and ultrasound. After this process, the encapsulation process was performed by ionotropic gelation using corn starch, sodium alginate, and *Weissella confusa* exopolysaccharide as wall material. The encapsulates produced presented a moisture content between 7.10 and 10.45% (w.b), a solubility of 53.06 ± 0.54%, and a wettability of 31.46 ± 2.02 s. The total phenolic content (TPC), antioxidant capacity of DPPH, and ABTS of the encapsulates were also determined, finding 232.55 ± 2.07 mg GAE/g d.m for TPC, 45.64 ± 0.9 µm Trolox/mg GAE for DPPH, and 51.69 ± 1.08 µm Trolox/mg GAE for ABTS. Additionally, ultrahigh performance liquid chromatography (UHPLC) analysis allowed us to identify and quantify six bioactive compounds: rosmarinic acid, caffeic acid, p-coumaric acid, ferulic acid, gallic acid, and quercetin. According to the above, using ionotropic gelation, it was possible to obtain microencapsulates containing bioactive compounds from pineapple peel extracts, which may have applications in the development of functional foods.

## 1. Introduction

Pineapple is a tropical fruit consumed around the world for its pleasant aroma and flavor, and it can be consumed fresh, in juice, jams, canned, and dehydrated, among other presentations [1,2]. During processing, a large amount of waste is generated, corresponding to 45–65%, with the peel being the largest proportion [3]. The disposal of these by-products represents a problem, since they are prone to contamination and can cause serious environmental problems; thus, it is necessary to generate alternatives that allow their use [4]. Some research has reported that pineapple peel has a great nutritional value, containing sugars, cellulose, hemicellulose, organic acids (quinic acid, citric, and malic), lignin, pectin, enzymes, and antioxidants such as vitamin A, C, and flavonoids [5,6]. According to the information reported, these compounds have a wide field of application, for example, in the food industry with preservative products, the replacement of synthetic antioxidants, or in the incorporation of functional products [7,8]. An important step in obtaining these compounds is the extraction method used. In the last few years, environmentally friendly extraction technologies have been used for the extraction of these compounds from waste, which provide high yields, lower solvent quantities, and reduced times [9].

Although the extraction methods employed provide extracts with good antioxidant activity, phenolic compounds oxidize easily and are sensitive to heat and light, with its application in the industry difficult to carry out [10,11]. Encapsulation is a suitable method to protect bioactive compounds, such as antioxidants, vitamins, and essential oils, against adverse factors [12,13]. Depending on the size of the capsule, they can be classified as microencapsulated or nano-encapsulated. In addition, the main advantage of this process is the controlled release of the compounds because it avoids their deterioration during digestion and delays their absorption during their passage through the gastrointestinal tract [14]. There are several methods for this process (spray drying, freeze-drying, emulsion system, coacervation, etc.), one of them being the extrusion technique. Extrusion is characterized as being a cost-saving one, since it eliminates the use of organic solvents and does not use high temperatures, which represents a great advantage to encapsulate bioactive compounds without generating thermal degradation. The size of the capsules obtained with this method ranges between 400 and 2000 µm [15,16].

Sodium alginate is a heteropolysaccharide extracted especially from brown marine algae that has multiple applications, one of the main ones being in the food industry, where it is used as an emulsifier, thickener, stabilizer, or gelling or texturizing agent; it has a great ionic gelling ability and high viscosity, this characteristic being an advantage for the retention of volatile compounds [17,18]; likewise, it is highly adaptable and has high cell affinity, different levels of degradability, and high mechanical resistance, among other advantages [19]. Corn starch is a wall material composed of amylose and amylopectin; it has been widely used to encapsulate various functional compounds. In addition, it has applications in the areas of food, pharmacy, chemistry, biotechnology, etc. Due to its abundance and biocompatibility, it is often used in encapsulation processes [20,21] because it has many of the properties necessary to carry out an optimal encapsulation process, such as low viscosity with a high solids content and high solubility [22,23].

Exopolysaccharides (EPS) have a high molecular weight and long-chain polymers, which are secreted externally into the microbial cell. In recent years, research on microbial polysaccharides has increased as they have advantages over those extracted from animals, algae, and plants [24,25]. For example, in the food industry, they can be used as viscosifiers, texturizers, emulsifiers, and syneresis-reducing agents [26,27]. The bacterial exopolysaccharide used in this study was obtained from *Weissella confusa*, which is a lactic acid bacterium. This EPS is characterized by its solubility and water retention capacity, which is influenced by its protein content (9.24 mg/L), and this property allows the polysaccharide to retain water without releasing the liquid [28,29]. The studies that have been carried out concerning the encapsulation of pineapple peel extracts have used spray drying, showing little retention of these compounds, and using maltodextrin, arabic gum, and inulin, among others, as wall materials. Therefore, this study focused on using extrusion technology with the objective of evaluating using room temperature during the encapsulation process and low temperatures (40 °C) during the drying process of the encapsulates, which would allow a higher retention of the bioactive compounds of interest. This study also sought to evaluate if the integration of the wall materials (sodium alginate, waxy starch, and *Weisiella confusa* exopolysaccharide) was appropriate to preserve the components and whether it offered improved properties regarding good solubility, wettability, morphology, etc.

## 2. Materials and Methods

### 2.1. Materials

Pineapple peels were supplied by a supermarket located in Cali, Colombia; the fruit was at maturity grade three according to the Colombian technical standard (NTC 729). The reagents used in item 2.2 were sodium hydroxide tablets (Merck, Darmstadt, Germany) and ethanol (Panreac, Barcelona, Spain).

The wall materials used in the research were waxy starch provided by Ingredion Colombia S.A., sodium alginate (Cape Crystal Brands, BJ, USA), and exopolysaccharide produced by the bacterium *Weissella confusa*. The material that produced exopolysaccharide was MRS agar (Scharlau, Barcelona-Spain), and the preparation of the culture media was carried out according to the specifications provided by the manufacturer. The media were stored at 5 °C until use. The *W. confusa* strain (CPQBA 1504-17 DRM 01) is an exopolysaccharide-producing lactic acid bacterium. This microorganism was obtained from the strain collection of the Microbiology and Applied Biotechnology Laboratory (MIBIA) of the Marine Biology section of the Faculty of Exact Sciences of the Universidad del Valle. It was activated in MRS broth and incubated at 37 °C for 24 h [30].

To produce exopolysaccharide (EPS), the method of González [31] was applied. The *W. confusa* strain was activated in MRS, and its purity was verified by performing Gram staining. It was then seeded on enriched medium (sugar, soy milk, buttermilk, and bacteriological agar), prepared according to the formulation determined by Gonzalez [31], and allowed to incubate for 36 h at 25 °C. The EPS produced was removed from the surface of the medium, spread on plastic sheets, and dried in an oven at 60 °C for 48 h. Subsequently, the dried polysaccharide was obtained, then ground, and sieved in a 300-micron mesh (Fisher Scientific, Waltham, MA, USA) sieve to obtain fine particles that were packed in low-density polyethylene bags with a hermetic seal and stored at 25 °C in a desiccator.

In the total phenolic content, the analytical grade of ethanol (Sigma-Aldrich, St. Louis, MO, USA) was used as well as Folin–Ciocalteau reagent (Panreac, Spain), sodium carbonate (Merck, Darmstadt, Germany), gallic acid (Sigma-Aldrich, St. Louis, MO, USA); for the antioxidant activity, 2,2-Diphenyl-1-Picrylhydrazyl reagent (Merck, Germany), 2,2′-Azino-bis(3-Ethylbenzothiazoline-6-Sulfonic Acid (Merck, Germany), and Trolox reagent (Sigma-Aldrich, St. Louis, MO, USA) were used.

Some standards derived from xanthines, catechins, flavonoids, and anthocyanins from Sigma-Aldrich were used as reference standards in the UHPLC analysis. Finally, for microbiological tests, culture medium potato dextrose agar (PDA, Merck), plate count agar (PCA, Scharlau, Dornstetten, Germany), MRS agar (Sharlau, Dornstetten, Germany) with aniline blue, and violet red bile agar (VRBA, Sharlau, Dornstetten, Germany) were used.

### 2.2. Pineapple Peels and Obtaining the Extract

The peels were washed, cut into 5 ± 1 mm^2^ squares, and dried at 50 °C for 24 h. These were then placed in aluminum trays, where these were inoculated with the fungus *Rhizopus oryzae* (1.8 × 10^7^ spores/g) to ferment the pineapple peels and increase the bioactivity of phenolic compounds present. The moisture content was adjusted to 85% (wet basis), with a pH of 5.5 with a 0.1 N sodium hydroxide solution, and they were incubated at 37.3 °C for 24 h. Subsequently, an ultrasonic bath (Transsonic 460/H, Elma, Singen, Germany) with a frequency of 35 kHz was used for the extraction process. To determine the characteristics of the ultrasound-assisted extraction process, a central composite design (CCD) was used, finding that the optimized conditions were 30 min, 58% ethanol, and 62.5 °C, parameters which were defined in previous research [30]. The samples were then centrifuged at 4000 rpm for 30 min (Hettich^®^ ROTOFIX 32A, Kirchlengern, Germany) and filtered on 0.45 µm cellulose nitrate filters (Sartorious stedim biotech). Afterwards, ethanol was evaporated from the supernatant using a rotary evaporator (Rotary evaporator IKA RV-10, Baden-Württemberg, Germany), the final solution (extract) was stored in amber vials until further use.

#### 2.2.1. Total Soluble Solids, pH, Color and Moisture Content

A digital refractometer (RX-7000X, Atago, Japan) was used for the measurement of soluble solids expressed as °Brix. A pH meter (Orion 710-A, Sigma Aldrich, Burlington, MA, USA) was used to perform the pH measurement. To determine the color coordinates L*, a* y b* (CIELAB), a colorimeter was used (ColorFlex EZ, Hunterlab, VA, USA), directly measuring the extract as described by [32]. Moisture determination was performed by drying in an oven (Binder, New York, NY, USA) at 105 °C until a constant mass was reached.

#### 2.2.2. Microencapsulation Process

The microencapsulation of the extracts was performed following the extrusion method used by Turhan et al. [33] and Shaharuddin and Muhamad [34] with some modifications. Sodium alginate (Cape Crystal Brands, Summit, NJ, USA), corn starch (Waxy), and EPS were used as encapsulating materials to prepare the microcapsules. The percentages used for each component were 73% starch, 20% sodium alginate, and 7% exopolysaccharide. The mixture of alginate and EPS in the solution was prepared and sterilized in an autoclave (Sterilmatic^®^, Market Forget/STM-E, Rockaway, NJ, USA) at 121 °C for 15 min, while the starch as well as the elements used for encapsulation were exposed to UV light in a laminar flow cabinet (C4/FLC85, Yumbo, Colombia) for 60 min. Likewise, to prepare the CaCl_2_ (Agenquímicos Ltd., Cali, Colombia) solution with a concentration of 3% *w*/*v*, distilled water was sterilized in an autoclave (121 °C), and the calcium chloride was exposed to UV light for the same time as the EPS and alginate mixture. 

Subsequently, the mixture of alginate and EPS was allowed to cool to 27 °C for the addition of starch and extract. An airbrush (Discover^®^ 600 cc, Bogotá, Colombia) with a nozzle diameter of 0.5 mm was used for microencapsulation to achieve a more homogeneous size in the formation of droplets due to the pressure experienced by the capsules. The ratio of extract to wall material used was (1:3 *w*/*w*). Subsequently, the mixture of encapsulating material with the added extract was passed through the airbrush into the calcium chloride solution in a 1:1 *v*/*v* ratio. Then, the contents were agitated in a shaking plate (Berchmark Scientific/H376O-H, Sayreville, NJ, USA) at 700 rpm for 15 min. Subsequently, the obtained mixture was centrifuged at 4000 rpm for 15 min. Afterwards, the supernatant was discarded, and the precipitate was poured into Petri dishes to proceed with drying in an oven at 40 °C for 3 h. The dried samples were subjected to a size reduction process using an analytical mill (IKA A11 basic, Württemberg, Germany) for a time of 5 min. Figure 1 shows the procedure for microencapsulation.

#### 2.2.3. Process Yield

The yield of the encapsulation process was determined by the mass ratio of solids present in the microcapsules to solids present in the extract, as presented in Equation (1) [35]:(1)$$\%Y=\frac{W_{encapsulates}\left(1-X_{w}{}_{encapsulates}\right)}{W_{extract}\left(1-X_{w}{}_{extract}\right)}\cdot 100$$
where *Y* is the yield of the encapsulates, $W_{encapsulates}$ is the mass (g) of the encapsulate obtained after the drying process (contain the pineapple extract and the wall materials), $W_{extract}\;$ is the mass (g) of the extract entering the extrusion process (does not contain the wall materials), and $X_{w}{}_{encapsulates}$ and $X_{w}{}_{extract}$ are the mass fractions of the moisture content on the wet basis of the encapsulates and extract.

### 2.3. Microparticle Analysis

#### 2.3.1. Particle Morphology and Size Distribution

The morphology of the encapsulates was determined by scanning electron microscopy (SEM), and the samples were placed on a JFC-1600 Auto-Fine Coater platinum-coated holder. Six runs were performed using 2 g per sample. The images of the microencapsulates were observed with the Phenom Pro-X SEM equipment at 1500 magnification, applying an accelerating voltage of 15 kV and backscattered electron detector. Sixteen photographs were taken, from which the mean size and the respective deviation were established.

#### 2.3.2. Moisture Content

For the determination of moisture content, the methodology reported in AOAC 934.06 (AOAC, 2002) used by Beltran et al. (2010) [36] was used. The samples were dried at 105 °C to constant weight to determine the moisture content. This measurement was performed in triplicate.

#### 2.3.3. Wettability and Solubility

The methodology of A-Sun et al. [37] was used to determine wettability. For this test, 50 mg of the encapsulate was deposited in 20 mL of water, and the time in which it was completely immersed without shaking was determined. It was performed at 28 °C.

Solubility was realized according to the methodology of Ee et al. (2014) [38] with some modifications. In 10 mL of water, 0.33 g of encapsulate was dispersed, a magnetic stirrer was used to aid dissolution, and this process was carried out for 30 min at 700 rpm. Then, it was centrifuged for 30 min at 5000 rpm. The supernatant was placed in Petri dishes and dried for 4 h at 60 °C. To determine the solubility of the encapsulates, the ratio of the initial mass of the added encapsulates to the mass of the dried supernatant was used.

#### 2.3.4. Bulk Density and Tapped Density

The methodology described by Fernandes et al. [39] was used to determine the bulk density ($\rho_{b}$) and tapped density ($\rho_{t}$). In a 5 mL test tube (previously weighed), the encapsulate was deposited until completing 2 mL. This weight was registered, and the bulk density was determined (mass/volume (g/mL)).

In a 5 mL test tube, 300 mg of encapsulate was deposited and subsequently tapped manually about 100 times, and the volume change in the measurements was registered. Once the mass and apparent volume were obtained, the tapped density was determined (*m*/*v* (g/cm^3^)).

#### 2.3.5. Flowability and Cohesiveness

The Carr index (CI) and Hausner ratio (HR) were used to determine the flowability and cohesiveness of the encapsulates, making use of the bulk and tapped density, using the following equations [4,40]:(2)$$CI=\frac{\left(\rho_{t}-\rho_{b}\right)}{\rho_{t}}*100$$
(3)$$HR=\frac{\rho_{t}}{\rho_{b}}$$

To determine the flowability and cohesiveness properties of the encapsulate, the values reported by [40] were considered. A Carr index <15 indicated very good flowability, while values >45 were very bad. For the Hausner ratio, values <1.2 indicated low cohesiveness, and values above 1.4 indicated high cohesiveness.

### 2.4. Evaluation of Phenolic Content and Antioxidant Activity of the Encapsulated Compounds

#### 2.4.1. Preparation of the Extracts from the Microcapsules

To obtain the extracts from the encapsulates, the methodology reported by Rocha et al. [41] was utilized with few modifications. In 10 mL of water, 200 mg of encapsulate was dispersed and was dissolved with an ultraturrax (IKA Labortechnik, T25 Basic, Staufen, Germany) for 3 min at 13,500 rpm. This suspension was left in the dark for 2 h and subsequently centrifuged for 15 min at 5000 rpm; the supernatant was used for further analysis.

#### Total Phenolic Content (TPC)

The methodology described by Gutierrez et al. [42] was used to quantify the total phenolic content. Briefly, 500 μL of the sample was taken and deposited in a test tube, 750 μL of Folin–Ciocalteau reagent was included, and the mixture was shaken and allowed to stand for 10 min. Subsequently, 750 μL of sodium carbonate was inserted and again allowed to stand for 10 min. The sample was read at 760 nm in a UNICO^®^ 2100 spectrophotometer; concentrations of 0–100 ppm gallic acid were used for the calibration curve.

The following equation, described by Escobar-Avello et al. (2021) [43], was considered to determine the encapsulation efficiency:(4)$$\%EE=\frac{TPC\;extracts-TPC\;encapsulates}{TPC\;extracts}*100$$
where TPC extracts is the phenolic content of the extracts, and TPC encapsulates is the phenolic content of the encapsulates obtained.

#### Antioxidant Activity by DPPH (2,2-Diphenyl-1-picrylhydrazyl)

The methodology of Ballesteros et al. [44] was used to determine the antioxidant capacity by DPPH. Briefly, 2900 μL of DPPH solution was added to 100 μL of the extract, mixed, and left in the dark for 30 min. Subsequently, the reading at 517 nm was performed in a spectrophotometer (Model 2100 Unico^®^, Princeton, NJ, USA). Equation (5) was used to determine the percentage of inhibition:(5)$$RSA\;\left(\%\right)=\frac{A_{c}-A_{s}}{A_{c}}*100$$
where *A_c_* is the absorbance of the control sample (DPPH-methanol), and *A_s_* is the absorbance of the sample (extract).

#### Antioxidant Activity by ABTS (2,2′-Azino-bis(3-ethylbenzothiazoline-6-sulfonic acid)

The methodology of Mesa et al. [45] was used to determine the antioxidant capacity by ABTS. First, the ABTS+ radical was prepared using potassium persulfate, the oxidation reaction was carried out for 16 h in the dark, and 80% methanol was included to adjust the absorbance of the ABTS solution to 0.70 ± 0.02 at a wavelength of 734 nm. To quantify the samples, 100 μL of the extract was used by adding 1800 μL of the ABTS⁺ solution. Samples were left in the dark for 30 min and read in a Model 2100 spectrophotometer (Unico^®^) at 734 nm. Trolox was used as the standard for the calibration curve. To determine the radical scavenging activity, Equation (6) was used:(6)$$RSA\;\left(\%\right)=\frac{A_{0}-A_{t}}{A_{0}}*100$$
where *A*_0_ is the absorbance of the sample at the initial time, and *A_t_* is the absorbance of the sample at thirty minutes.

### 2.5. Identification of Bioactive Compounds by High Performance Liquid Chromatography (UHPLC)

This analysis was developed following the methodology described by Polanía, Londoño, Ramírez, and Bolívar (2022) [30]. Briefly, the samples analyzed were prepared by dissolving the sample in a mixture of methanol: 0.2% water in formic acid (1:1), mixing (5 min), sonication (5 min), filtering, and subsequent injection into the chromatographic equipment. An ultra-high efficiency liquid chromatograph (UHPLC), Dionex Ultimate 3000 (Thermo Scientific, Sunnyvale, CA, USA), equipped with a binary gradient pump (HP G3400RS), an automatic sample injector (WPS 300TRS) and a thermostated column unit (TCC 3000), was used.

### 2.6. Fourier Transform Infrared Spectroscopy (F-TIR)

This analysis was performed using a Fourier Transform Infrared Spectrometer Jasco (FT/IR-4100), and the data were obtained under the scanning range of 350–7800 cm^−1^ with a resolution of 0.7 cm^−1^. A tablet was prepared, which was composed of 100 mg of potassium bromide (KBr) with 2 mg of sample; a spatula was used to integrate both components. The salt was previously ground in an agate mortar. Finally, the samples were read on the equipment. All measurements were performed threefold.

### 2.7. Microbiologic Analysis

For the microbiological analysis, 1 g of the encapsulate was taken and homogenized in 9 mL of peptonized water with the help of a Stomacher (MiniMix 100PCC Lab Blender, FisherBrand, Madrid, Spain) for 15 min. From this mixture, dilutions up to 10^−6^ were made, taking 100 µL, which were inoculated in Petri dishes with culture medium potato dextrose agar (PDA, Merck), plate count agar (PCA, Scharlau), MRS agar (Sharlau) with aniline blue, and violet red bile agar (VRBA, Sharlau), and these were incubated at 30 °C for 24 h, in triplicate. Microbiological counting was performed considering the method recommended by the Colombian Technical Standard (NTC 926, 2016).

### 2.8. Statistical Analysis

All the experiments were conducted as three independent replicates, and results were expressed as mean ± SD. 

## 3. Results and Discussion

### 3.1. Total Soluble Solids, pH, Color, Moisture Content, and Antioxidant Activity

Table 1 shows the main physicochemical properties of the extract obtained and passed through the rotary evaporator.

It is possible to observe that this extract presented a high moisture content (95.90% ± 0.05) and an acid pH (3.85 ± 0.02). This pH value could be attributed to the presence of organic acids in the extract such as: caffeic acid, vanillic acid, p-coumaric acid, ferulic acid, rosmarinic acid, and gallic acid [30]. These parameters were similar to that reported by Selani et al. (2014) [46], who worked with by-products of honey-gold pineapple (peel and pomace), recording a value of 3.86. It also coincides with those described by Lourenço et al. (2021) [9], who found values of 4.02 and 3.70 in extracts obtained with water and 80% ethanol in Sweet Gold variety pineapple peels, and regarding the moisture content, the results were similar with 95.47 and 93.72, respectively.

The lightness parameter indicates whether a sample tends to be white (100) or black (0). According to the results, it is possible to infer that the sample is dark (24.65 ± 0.13) compared to the control sample (without extract) (47.06 ± 0.84) (Figure 2), which can be attributed to the temperature and exposure time that was applied during the solvent evaporation process (60 °C, 2 h) using the rotary evaporator. This behavior was similar to that reported by Rittichai et al. [47], who evaluated different evaporation methods on pineapple juice concentrates, reporting that the decrease in this parameter is directly related to the increase in temperature and processing time, leading to an increase in browning reactions.

Regarding soluble solids, these increases after ripening and changing depend on the maturity stage of the fruit. Specifically, ranges from 8.6 to 18.0 °Brix have been reported for the MD2 variety [47,48], with values similar to the one found in this study (10.13 ± 0.06). It is also important to highlight that the MD2 variety presented a higher value of soluble solids compared to the Moris and N36 variety at all stages of maturity, presenting a sweeter flavor from an early stage of maturation [49].

Before encapsulation, the total phenolic content of the extract was 882.71 mg GAE/g d.m, while the antioxidant activity of DPPH was 111.55 µm Trolox/g dry extract, and for ABTS, 325.23 µm Trolox/g dry extract was obtained. The values found were higher than those reported by Samarakoon and Rajapakse (2022) [50], who obtained methanolic extract of a pineapple peel to analyze its potential as a natural source in sun protection cosmeceutical products, reporting 0.208 mg GAE/g dry weight. Likewise, it was higher than that indicated by Valdés García et al. (2021) [51], who evaluated the antioxidant activity in pineapple peels using methanol as a solvent, showing values between 15 and 41 mg GAE/100 g sample. Regarding DPPH, the values were similar, with 138–243 µm Trolox/100 g sample, and for ABTS, 124–158 µm Trolox/100 g sample. However, it is important to emphasize that it is not possible to compare these results with the literature because these compounds change depending on the state of the sample, extraction conditions, and the extraction process. During fermentation, enzymes such as xylanase, β-glucosidase, amylases, and esterases, among others, are produced that allow the release of phenolic compounds present in the sample [52]. On the other hand, during ultrasounds, mechanical and acoustic effects are generated that cause bubbles to break in the liquid extraction medium. These bubbles implode and break down the cell walls of the peel, increasing the amount of free polyphenols [53].

### 3.2. Process Yield

This parameter is considered one of the key attributes to establish the cost and efficiency of the encapsulation process [54]. The yield obtained (74.87 ± 1.07%) was mainly due to the two wall materials used in greater proportion: corn starch and sodium alginate (73% and 20%, respectively). Corn starch presents a low viscosity with a high solids content and high solubility [22]; on the other hand, sodium alginate gives the encapsulates good mechanical strength and elasticity [55]. This yield is high if compared with the results reported by Mousavi Kalajahi and Ghandih [56], who encapsulated nettle extract (*Urtica dioica* L.) by spray drying, employing maltodextrin as a wall material and obtaining ranges between 9.72 and 39.05%. This can be associated with the fact that a spray dryer employs high inlet air temperatures, which affect the phenolic compounds present, while the extrusion process is realized at room temperature. Likewise, Braga et al. [35], who used spray drying to encapsulate pineapple and mint juice, obtained yields of 34.40, 43.66, and 62.92% when encapsulating with 0, 3, and 15% maltodextrin, showing that better yields were obtained with an increased quantity of maltodextrin. However, the results obtained in this study were higher because the temperature used during extrusion encapsulation was 28 °C (ambient), while the conditions established by [35] were a feed flow rate of 3.4 mL/min, an inlet air temperature of 150 °C, and a maltodextrin concentration of 16.9 g/100 g. This high temperature affects the functional properties of the nettle extract encapsulated.

### 3.3. Microencapsulates

#### 3.3.1. Particle Morphology and Size Distribution

Figure 3 shows the external structure and morphology of the encapsulates. It is possible to observe that irregular shapes are present. The differences in the shapes of the samples could have arisen due to changes in conductivity, solution concentration, and the encapsulated substance [57], since, for example, when sodium alginate makes contact with Ca^2+^ ions in the hardener solution, cross-linking reactions occur, creating networks that influence the morphology of the particles [58]. Similarly, the functional groups present in the encapsulated extract can interact with the components present in the wall materials and thus affect the structure of the encapsulates [59].

It is possible to appreciate that the encapsulates presented agglomeration in some areas. The encapsulates within the agglomerates showed average diameters ranging from 5.25 to 11.6 µm (Figure 4). This phenomenon could be due to the presence in a greater proportion of waxy starch as wall material, which can give rise to these agglomerations due to its composition; corn amylopectin has a low amount of short chains and a greater amount of long unit chains, which makes it necessary to apply high temperatures to dissociate them completely; given that the process was carried out at room temperature (28 °C), this favored greater agglomeration [60,61].

However, this type of agglomeration provides greater stability to the encapsulated extracts due to the barrier that the external particles can confer to the internal ones. This same behavior was observed by Lourenço et al. (2021) [9], who performed the spray-drying encapsulation of pineapple peel extracts using maltodextrin as the encapsulating material. No imperfections were observed on the surface of the encapsulates, such as cracks or crevices, and it showed great protection and withholding of the extract molecules [62]. These particle agglomerations were also mentioned in other works, such as encapsulates obtained by the spray dryer of aqueous extracts of Andes Berry (*Rubus glaucus Benth*), employing cassava starch and maltodextrin as wall materials [63] and pineapple juice powder encapsulated with 20% maltodextrin [64]. This stability can also be attributed to the interaction between waxy starch and sodium alginate because they are the main ingredients of the wall material (73 and 20%). Between these two components, a complex matrix network was formed because the hydroxyl groups of the starch interact with the carboxyl groups of the sodium alginate through hydrogen bonds. These interactions are strong; therefore, a compact structure is formed [65].

#### 3.3.2. Bulk Density, Tapped Density, Flowability, and Cohesiveness

Bulk density is a parameter that must be considered for packaging, transportation, and storage calculations, since these variables are necessary to calculate the amount of product that will fit into a container. High bulk density values result in enhanced packaging, and low values enhance the risk of product oxidation due to air within the powder. Bulk density values for the encapsulates ranged from 0.354 to 0.593 g/cm^3^. These values were influenced by the particle size distribution, i.e., powders of smaller particle size had higher bulk density. These values were similar (0.18–0.30 g/cm^3^) to those reported by Lourenco et al. [4], who performed the encapsulation of hydro-alcoholic extracts of pineapple peels by spray drying using maltodextrin, inulin, and gum arabic as wall materials. The values for tapped density ranged from 0.379 to 0.659 g/cm^3^, which were close to those obtained in the research of [4] (0.24–0.40 g/cm^3^). The tapped density values were higher than the bulk density. This behavior could be described by the formation of many bonds between particles, which generates a compact encapsulation with higher density. This trend was also reported by Ribeiro et al. [66] in their research on the flow behavior of cocoa pulp powder, using maltodextrin as wall material.

The flowability and cohesion of powdered materials are properties that can be related by the Hausner ratio (HR) and Carr index (CI). The former was used to determine the cohesion of the product, i.e., low values indicated encapsulations with good flowability. The Carr index was used to evaluate the compressibility of powdered products, i.e., high values reflect high compressibility and low flowability. The results obtained for the encapsulates were a Carr index of 8.34 ± 1.92% and a Hausner ratio of 1.10 ± 0.023. These results were in accordance with the bulk tapped density values. In general, the higher the cohesiveness, the greater the collapse of the powder observed on tapping [67]. According to these results, the encapsulates presented excellent flowability (<15%) and low cohesiveness (<1.2). Often, the flow properties are correlated with the properties of the wall material used in the process; particularly, in this study, the principal wall material was Waxy corn starch, which has shown fair and passable flow properties in other studies [68,69]. Based on Zhang et al. [70], encapsulated products with optimal flow properties are ideally suited for processing and handling operations.

#### 3.3.3. Moisture Content and Water Activity

The encapsulates presented a moisture content between 7.10 and 10.45%, which can be attributed to the different packing density arrangement of the exopolysaccharide, which directly influences the affinity for water in polymeric components [71], and because the drying process was carried out for a short time (3 h) to preserve the bioactive compounds present in the extract. The range was close to those reported in other investigations where spray-dried encapsulation was performed in pineapple juice with maltodextrin as the wall material [72]. Acerola pomace extract with maltodextrin and cashew gum [73] and pineapple peel extracts employing maltodextrin, inulin, and gum arabic, where values ranged from 1.87 to 6.29%, were determined [4]. This range of moisture content was similar to that related by Martinić et al. (2022) [74], who encapsulated dandelion (*Taraxacum officinale* L.) leaf extract by spray drying, reporting that when they used alginate as a wall material, they obtained 7.3 ± 0.4% moisture content using an inlet temperature of 130 °C and an outlet temperature of 66 °C.

The water activity of the encapsulates was 0.57 ± 0.03, a value that guarantees the microbiological stability of the product, since aw <0.60 [75]. Additionally, the values determined were below the maximum acceptable limit to avoid the degradation of food products, as reported in the Labuza diagram [76]. These results are higher than those reported by Bustamante et al. (2017) [77], who encapsulated pomegranate peel and seed oil by spray drying, using waxy corn starch as encapsulant material, reporting water activity ranges between 0.22 and 0.28. These results could be attributed to the air inlet temperatures that ranged from 150 to 205 °C. Tan et al. [78] also reported values of 0.33 ± 0.01 in encapsulations of bitter melon aqueous extract by spray drying with maltodextrin (MD) and arabic gum (GA) as wall materials. In another study by Quek et al. (2007) [79] who performed spray drying with watermelon powders using two different concentrations of maltodextrin (3% and 5%) as the encapsulating material. They evaluated different temperatures, and ranges between 0.20 and 0.29 were determined for water activity. The researchers reported that increasing the inlet temperature increased the heat transfer to the particles, generating a significant driving force that enhanced water evaporation. Although the a_w_ of the encapsulates obtained in this work was not so low, it was below the limit considered as critical. In addition, the storage conditions also performed an important role in this matter, i.e., the encapsulates should be properly stored in an airtight container and kept in a cool and dry place.

#### 3.3.4. Wettability and Solubility

Solubility is a significant property that directly affects the availability of encapsulated compounds when incorporated into a food matrix [80]. Additionally, encapsulates with low solubility can pose a problem during processing. The solubility values obtained were in the range of 53.44 ± 0.54%, indicating that they are partially soluble in water. This solubility can be attributed to the wall materials employed; corn starch was the main component. This biopolymer is widely used because it is inexpensive, readily available, and has optimal viscosity and water retention properties [81]. Likewise, it has adequate resistance to digestion, which is ideal for encapsulation processes [82]. This value was higher than that described by Kobo et al. (2022) [83], who encapsulated passion fruit (*Passiflora edulis Sims*) peel extracts using spray drying and different wall materials, with the lowest solubility reported when using waxy starch (30.67 ± 5.46%). It is possible that the solubility is correlated with the particle size of the powders. There is a correlation between particle size and hydration capacity, i.e., the smaller the particle size, the greater the hydration capacity due to the increase in surface area [84]. Although the encapsulates had small particle sizes (8.43 ± 3.18 µm), this solubility percentage was affected by the agglomeration of the particles. This pattern was also observed by Nthimole et al. (2022) [85], who encapsulated raspberry juice and employed maltodextrin, gum arabic, and waxy starch as encapsulating agents, obtaining solubilities of 60.83 ± 0.08%, 60.25 ± 0.14%, and 54.52 ± 0.03%, respectively. The authors mentioned that despite obtaining the smallest particle size with the waxy starch encapsulates, this solubility was affected by the agglomerations of the powders.

For its part, sodium alginate is a biopolymer with high availability and biodegradability that is biocompatible with other wall materials [86]. It has been reported that employing alginate as a wall material together with other polymers can increase the mechanical and chemical stability, thus improving the effectiveness of the encapsulation system [87]. In other words, the combination of sodium alginate with starch is a porous matrix. However, when the exopolysaccharide is added, a semi-permeable membrane is formed around the matrix, which limits the diffusion of the encapsulated extract [28,88], which was desirable in this case to preserve the pineapple peel extract within the wall material. The values obtained for solubility (53.44 ± 0.54%) were close to those reported by Lourenço et al. (2020) [4], who also encapsulated pineapple peel extracts but employed maltodextrin, inulin, and gum arabic as wall materials, reporting ranges between 62 and 75%. Comparable results have been found for grape peel extracts encapsulated by spray drying, using waxy starch, gum arabic, and maltodextrin as wall materials, finding that depending on the type of material used and the concentration, its solubility is affected. These authors reported ranges between 28 and 52% for waxy starch, 64 and 94% when gum arabic was used, and 68 and 92% when maltodextrin was used [89]. It is possible to observe that when waxy starch is used as a wall material, the solubility of the encapsulate decreases due to the structural characteristics of starch. This behavior is corroborated by Yousefi et al. [89] and other authors that mentioned that when starch is used in high concentrations, the solubility of the encapsulates decreases. However, the result obtained in this research indicated a good solubility of the product since it is a mixture of waxy starch, sodium alginate, and *Weissella confusa* exopolysaccharide.

The capacity of encapsulates to rehydrate is known as wettability. It is an important property in products that are easy to prepare (ready to dilute). Encapsulates are considered wettable if the wetting time is less than 60 s and non-wettable if it is greater than 120 s [57]. The encapsulates presented a time of 31.46 ± 2.0 s, indicating that they are easy to reconstitute. This parameter is strongly influenced by the shape, particle size, humidity, and structure of the carrier agent [90]. In this sense, the wettability value obtained in this work can be clarified by the fact that the particle size was 8.43 ± 3.18 µm, the moisture content ranged between 7.10 and 10.45%, and the wall material that was present in greater proportion was waxy corn starch, which presents a good wettability and for this reason is used in soups, willows, and snacks [91]. In addition, the water penetration in the encapsulates was facilitated by their moisture content. The wetting of the surface of these materials by the liquid (usually water) is often the limiting step to reconstitute them [35]. 

This result obtained was lower than the one reported by Quoc (2020) [90], who performed the encapsulation of pineapple powder (*Ananas comosus Merr*.) with a spray dryer, evaluating maltodextrin and the combination with gum arabic as wall materials and obtaining wettability values of 87.33 ± 19.35 s and 1153. 67 ± 13.01 s, respectively. The author mentioned that the mixture of the wall materials in this study presented agglomerations, meaning the aggregation of dispersed materials to larger material units, which increased the wettability when the two wall materials were incorporated. Furthermore, the wettability value in this work was lower than that of Braga et al. [35], who evaluated different percentages of maltodextrin (0, 3 and 15%) to encapsulate by spray drying pineapple-mint juice. They obtained 261 s, 323.4 s, and 813.6 s, respectively, and concluded that the enhancement in maltodextrin concentration influenced a longer wetting time, while influencing the maltodextrin used (DE 10), since this product is less hydrolyzed. Thus, it showed less hydrophilic groups and less water bonds, which increase the wetting time [92].

#### 3.3.5. Total Phenolic Content and Antioxidant Activity by DPPH and ABTS

Before encapsulation, the total phenolic content of the extract was 882.71 mg GAE/g d.m, while the antioxidant capacity for DPPH was 111.55 µm Trolox/g dry extract, and for ABTS, 325.23 µm Trolox/g dry extract was obtained. After the encapsulation process, there was a decrease in the parameters of antioxidant activity and total phenolic content because the particles were exposed to a drying process at 40 °C for 3 h, which affected the bioactive compounds present in the extract, finally obtaining 232.55 mg GAE/g d.m for TPC, 25.64 µm Trolox/g m.s for DPPH, and 35.57 µm Trolox/g d.m for ABTS. These results evidenced that the combined process of solid-state fermentation together with ultrasound allowed for a significant increase in these compounds. During fermentation, a great number of hydrolyzing enzymes are produced, whose main function is to release phenolic compounds bound to the substrate. Additionally, ultrasound generates acoustic and mechanical effects that create bubbles that implode, disintegrating the cell wall of the peels and releasing the polyphenols [30]. 

The results of TPC and the antioxidant activity of encapsulates were higher than those reported by Lourenço et al. (2020) [4], who encapsulated by spray drying hydroalcoholic extracts of pineapple peels of the variety ‘Sweet Gold’ using maltodextrin, inulin, and gum arabic as wall materials. After the encapsulation process, they obtained ranges for total phenolic content from 3.42 to 4.82 mg GAE/g d.m and for antioxidant activity for DPPH ranges from 39.7 to 56.5 µmol Trolox/mg d.m. Comparing the results obtained with our research, it is possible to conclude that for TPC there was an increase of 22 times more, but with respect to the DPPH value, the results of Lourenço et al. were 1.8 times higher. Likewise, the results obtained were higher than those reported in the work of Lourenço et al. (2021) [9], who performed solid–liquid extraction on the ‘Sweet Golden’ variety of pineapple peels using 80% ethanol at room temperature (25 °C), obtaining 11.10 ± 0.01 mg GAE/g d.m and 91.79 ± 1.98 µmol Trolox/g d.m for DPPH. The authors reported that, after encapsulation by spray drying, they obtained 2.74 mg GAE/g d.m for total phenols and 10.49 ± 0.09 µmol Trolox/mg GAE for DPPH, observing that there was a significant reduction in both parameters. The drying process in both works had a negative impact on the antioxidant capacity, since this method generates chemical and structural changes that affect these properties. Compared to the results previously reported, this research was able to preserve in greater proportion the total phenolic content and the antioxidant capacity by DPPH. However, it is important to note that the variety used in this work was different (MD2) and that the authors mentioned did not report the maturity stage of the fruit. Although these values allow us to glimpse that the process of fermentation combined with ultrasound generates an enhancement in phenolic content and antioxidant capacity, the efficacy of these compounds obtained depends mainly on their stability in storage and processing. For this reason, the extrusion method would be a recommended technique to preserve these sensitive bioactive compounds.

Comparing the results of antioxidant activity, the DPPH and ABTS parameters evaluated were higher (61.97% and 80.03%) than those reported by Anuar et al. (2021) [93], who performed the encapsulation process by spray drying Kantan extract (*Etlingera Elatior*) and employed wall materials such as Hanjeli Flour, Modified Cassava Flour (MOCAF), and maltodextrin (5, 10, 15, 20, and 25%). The values reported for DPPH were between 38.14 and 44.39% and between 36.15 and 41.71% for ABTS. Therefore, the antioxidant activity was good despite being a process involving high temperatures (180 °C) because the wall materials protected the extract, which is highly sensitive to temperature. In the work conducted by Quoc (2020) [90], who encapsulated pineapple juice from the Tien Giang province (Vietnam) and evaluated different wall materials, such as maltodextrin (MD) and the combination of maltodextrin and gum arabic (MD-GA), the authors reported values for total phenolic content between 0.16± 0.01 and 2.64 ± 0.09 mg GAE/g d.w. These values were lower than those of the fresh material due to the degradation of the bioactive compounds by the high temperature of the process (160 °C). Considering this, the results of this research can be regarded as positive, given that it was possible to preserve the bioactive compounds in the encapsulates by means of a low process temperature (40 °C), a short drying time (3 h), and the combination of encapsulating agents that offer protection. It is relevant to highlight that the reported differences in the antioxidant capacity values can be associated with the composition of the encapsulated extracts, with the different types of wall materials, and with the different conditions in the drying processes applied.

#### 3.3.6. Encapsulation Efficiency

This parameter depends on many aspects, including the technique used for encapsulation, the size and solubility of the encapsulated compounds, the concentration of the components, and the affinity between the compounds and the wall materials [94]. The proportions used for the wall material were 73% waxy starch, 20% sodium alginate, and 7% exopolysaccharide. The results obtained showed that the process presented a good encapsulation efficiency (68.86 ± 1.75%), if compared, for example, with the work realized by Wongverawattanakul et al. (2022) [95], who used the extrusion technique to encapsulate polyphenols from *Mesona chinensis Benth* extract using sodium alginate as the wall material at 1.2, 1.5, and 1.8% *w*/*v*. The author reported an encapsulation efficiency that ranged between 41.1 ± 4.7 and 56.7 ± 3.4%, which was lower than the results of our research. This can be related to the fact that both the waxy starch and the exopolysaccharide were able to better preserve the extract and generate a good protection. The exopolysaccharide managed to create a semi-permeable layer around the alginate–starch matrix, which is porous, managing to limit the diffusion of the encapsulated extract, and the starch generated a resistant wall. It would be necessary to apply high temperatures to achieve its dissociation [28,60]. Likewise, in the research carried out by Lavelli and Sri Harsha (2019) [96], an encapsulation of 68% was determined, similar to the one found in this research. The authors used a spray drying process to encapsulate extracts obtained from grape peel, employing alginate hydrogels as wall material. The researchers pointed out that alginate has the potential to react with divalent cations, such as Ca^++^, to form a cross-linked gel structure that has the ability to hold several compounds in its interior. This coincides with the results in this research, since 2% calcium chloride was used as the hardening solution. In addition, being an anionic polymer with carboxylic terminal groups, alginate is a good mucoadhesive agent [97]. This efficiency was also higher when compared to the work conducted by Quoc (2020) [90], who encapsulated Golden pineapple extracts using a spray drying process with carrier agents such as maltodextrin and the combination of maltodextrin and gum arabic (MD-GA) with a MA/GA ratio of 70/30. Efficiencies of 51.65 ± 0.7% with maltodextrin and 47.31 ± 0.78% with the combination were found. The lower efficiency obtained is attributable to the fact that gum arabic has a short-branched chain structure and is strongly hydrophilic; thus, the powder readily adheres to the chamber wall, decreasing the efficiency.

#### 3.3.7. Compound Identification by UHPLC

Table 2 shows the results obtained for the UHPLC analysis performed on a sample of the encapsulate obtained under optimized conditions. In these microencapsulates, it was possible to quantify six compounds: caffeic acid, p-coumaric acid, ferulic acid, quercetin, rosmarinic acid, and gallic acid. These phytochemical compounds have been identified in pineapple peel extracts obtained through different extraction methods and using solvents such as ethanol, methanol, and hydro-ethanol combinations [4,9,49,97,98,99,100,101]. Achieving the identification and quantification of these compounds through this analysis allowed us to be sure that the encapsulation method used was effective in preserving the bioactive compounds present in the MD2 variety pineapple peel extract.

These identified compounds are similar to those reported by other authors. For example, Li et al. (2014) [99] identified four compounds in methanol extracts of pineapple peels: gallic acid, catechin, epicatechin, and ferulic acid. Two of these compounds coincide with those found in this work. Lubaina et al. (2020) [102], who evaluated different solvents for the Soxhlet extraction of phenolic compounds in pineapple peels of the *Mauritius* variety, identified and quantified twelve compounds: gallic acid, catechol, chlorogenic acid, caffeic acid, syringic acid, p-coumaric acid, ferulic acid, ellagic acid, myricetin, cinnamic acid, quercetin, kaempferol, and apigenin. Five of these coincide with those reported by our analysis; however, the values reported by Lubaina et al. referred to the extract and in this analysis were quantified in the encapsulates. Although, it is important to emphasize that these compounds presented a reduction after the encapsulation process, since the compounds present in the extract before encapsulation were: rosmarinic acid (0.412 µg/mg), caffeic acid (0.170 µg/mg), vanillic acid (0.063 µg/mg), coumaric acid (0.063 µg/mg), ferulic acid (0.051 µg/mg), quercetin (0.005 µg/mg), rutin (0.012 µg/mg), quercetin 3-glucoside (0.012 µg/mg), kaempferol 3-glucoside (0.0024 µg/mg), and gallic acid (0.0024 µg/mg). This shows that the applied temperature (40 °C) affected the amount of compounds initially present in the extract but achieved to preserve six of the ten compounds initially present.

The compound identified in the greatest quantity was rosmarinic acid, which has not been reported in the literature for this type of variety. This acid was first isolated from the leaves of *Rosmarinus officinalis*, and it belongs to the hydroxycinnamic group, although it has been reported in several plant species [103]. Research has been reported showing its potential as an antiallergic, antimicrobial, antidepressant, antidiabetic, and anti-inflammatory, and anticarcinogenic, nephroprotective, cardioprotective, hepatoprotective, and neuroprotective properties have also been associated with this compound, mainly due to its antioxidant potential [104,105].

#### 3.3.8. Fourier Transform Infrared Spectroscopy (F-TIR)

The encapsulation efficiency of phenolic compounds through ionotropic gelation can be estimated through F-TIR. Infrared spectroscopy analysis was performed to characterize the functional groups of the components that formed the microencapsulates. The spectra of extract, wall material, and encapsulates are presented in Figure 5a–c, respectively.

It is possible to observe some characteristic functional groups as is the case of O-H groups present in phenolic compounds, which are associated with the band of 3043 cm^−1^ (Mark 1); on the other hand, in the band of 1640 cm^−1^, the characteristic pattern of water strongly bound to starch was observed, and finally, the characteristic of stretching C-H associated with hydrogen atoms of the ring was found at a wavelength of 2926 cm^−1^ (around 906 cm^−1^ (Mark 2)) [106]. These characteristic wavelengths of the functional groups were used as references according to Fang et al. (2002) [107].

Particularly, in the extract, the peaks of 2977.55 cm^−1^ and 2894.63 cm^−1^ indicated the presence of alkanes. The signals at 2832.92 cm^−1^ correspond to hydrogen bond OH stretch), and 1648.84 cm^−1^ (C=O stretch) refers to the presence of carboxylic acid. The peak at 1452.14 cm^−1^ could be associated with (N=O stretch), and 1383.68 cm^−1^ together with 1325.82 cm^−1^ (N=O bend), which corresponds to the stretching and bending vibrations of the nitro group. The peaks between 1274.72 and 1085.73 cm^−1^ indicate the stretching vibrations of esters and ethers (C-O stretch). Absorption below 1000 cm^−1^ corresponds to the C-H swing vibration typical of cellulose present in pineapple peels, which is an anomeric vibration specific for β-glycosides [108]. The results obtained in this research were similar to those reported by Lubaina et al. (2019) [109], who analyzed pineapple peel extracts obtained through different solvents. Although it is important to mention that the transmittance values are not the same because of the influence of the extraction methodology used, the composition of the fruit, its state of maturity, and the variety are, however, very close to those reported. Other researchers have also reported similar values; for example, Mendez-Flores et al. (2018) [110] identified hydroxyl groups at 3235.69 cm^−1^, carboxylic acids (1702.49 cm^−1^), aromatic rings (1445.47 cm^−1^), and carbonyl groups (1080.80 cm^−1^), stating that they were the most important signals of functional groups of phenolic compounds present in rambutan peel of the Mexican variety. In other research realized by Essifi et al. (2021) [86], who evaluated the influence of sodium alginate concentration on the release of bioactive compounds, such as gallic acid and crocin, reported in the spectrum of gallic acid two bands attributed to the OH function (3268 and 3491 cm^−1^) and two other bands that characterized the C=O group (1605 and 1307 cm^−1^). They also attributed the band of 1021.12 cm^−1^ to the vibration of the benzene ring.

The ionic bond formed between the COO^−^ group of the alginate and the Ca^2+^ of the hardener solution during the formation of the microcapsules was observed both in the encapsulates and in the wall materials, and it was found at 2151.2 cm^−1^ and 2884.99 cm^−1^ [58]. The crosslinking that occurs during the capsule hardening process produces changes in the intensity of the asymmetry and symmetry of COO^−^ stretching bands [95,111]. The OH stretching in the wall and encapsulated materials changed to a higher wavenumber and increased the intensity if compared to the extract value (3367.1 cm^−1^). The wavelength in the encapsulated ones was higher (3427.85 cm^−1^). This can be explained by the intermolecular hydrogen bonds that were formed between the peel extract and the wall materials. A similar behavior was observed by Wongverawattanakul et al. [95], who observed this trend when comparing control beads and Mesona chinensis Benth extract -loaded alginate beads. Therefore, F-TIR allows for the identification of functional groups and their interactions in the materials studied, observing the presence of phenolic compounds in the extracts and the way in which the wall materials used behave during the capsule formation process.

#### 3.3.9. Microbiological Analysis

According to the results obtained for the microencapsulates, the counts of enteric microorganisms such as *Escherichia coli* and *Staphylococcus* were negative. Likewise, it was observed that there was no growth for molds and yeasts or lactic acid bacteria, indicating that the aseptic conditions used throughout the process were adequate to obtain a safe product. A comparison was made with the Colombian Technical Standard (NTC 926 of 2016) where the microbiological parameters for milling products and unmodified corn starch are specified, since the majority component in the formulation of the encapsulated products was corn starch. The results are presented in Table 3, and it is possible to appreciate the inoculated plates after 24 h in Figure 6.

a_w_ influences physicochemical parameters such as cell viability, enzymatic activity, non-enzymatic browning, and mainly microbiological parameters. Since the encapsulates presented values of 0.57 ± 0.03, this value was less than 0.6, which is appropriate to avoid the proliferation of microorganisms. In addition, this is an important parameter since it significantly influences the stability of the encapsulates during storage [80]. It is important to emphasize that the composition of the wall materials also influences the stability of the active components [112].

The null growth observed in all the plates could be attributed to the presence of some phenolic compounds present in the extract; the compound present in the highest proportion was rosmarinic acid. Rosmarinic acid antifungal mechanisms include the reduction of mitochondrial activity, alteration in membrane integrity, and the slight inhibition of protease production. The antimicrobial potential of rosmarinic acid has been confirmed towards eleven *Candida* strain inhibitions and thirteen bacterial pathogens (*Rothia mucilaginosa*, *Micrococcus luteus*, *Streptococcus agalactiae*, *Streptococcus anginosus*, *Streptococcus dysgalactiae*, *Streptococcus oralis*, *Streptococcus parasanguinis*, *Streptococcus pyogenes*, *Streptococcus salivarius*, *S. aureus*, *Staphylococcus hominis*, *Enterobacter cloacae*, and *Stenotrophomonas maltophilia*) [113].

## 4. Conclusions

Sodium alginate, waxy corn starch, and *Weisiella confusa* exopolysaccharide were analyzed as carrier agents to encapsulate a hydroalcoholic extract of MD2 pineapple peel obtained by the combined process of solid-state fermentation with ultrasound. The powders produced were characterized in their physicochemical properties, phenolic compound content, and antioxidant activity. The qualities present in the encapsulates can be attributed to the interaction between the wall materials used as well as their ratios. The wall materials and the drying temperature used (40 °C) allowed for the production of powders with adequate water activity content (0.57 ± 0.03), appropriate solubility (53.06 ± 0.54%), and wettability (31.46 ± 2.02 s), enabling them to be soluble in aqueous media. The morphology of the particles was irregular with a size from 5.25 to 11.6 µm; however, holes were not present on the surface, and the agglomerations observed offered greater stability in the encapsulated extracts. Regarding the FT-IR analysis, functional group characteristics of pineapple extract, such as alcohols, phenols, alkanes, alkenes, carboxylic acids, and nitro compounds, were identified. A new peak was spotted in the encapsulates, attributed to the interaction between sodium alginate and Ca^2+^ of the hardness solution. In the total phenolic content and antioxidant activity, 90.44 ± 2.07 mg GAE/g d.m. was obtained for TPC, 25.64 ± 0.9 µm Trolox/mg GAE for DPPH, and 51.69 ± 1.08 µm Trolox/mg GAE for ABTS, which show the viability of the method for preserving bioactive compounds. With respect to ultra-high performance liquid chromatography (UHPLC) analysis, it was possible to identify and quantify six bioactive compounds: rosmarinic acid, caffeic acid, p-coumaric acid, ferulic acid, gallic acid, and quercetin, which are characterized by their antioxidant properties that provide health benefits. Likewise, due to the antibacterial and antifungal properties evidenced in these compounds in the microbiological analysis performed, no type of microorganism grew, making it a safe product suitable for incorporation in the preparation of functional foods. It is recommended to carry out digestibility tests of the encapsulates to evaluate the absorption capacity of the compounds through the gastrointestinal tract over time.

## Figures and Tables

**Figure 1 foods-12-02943-f001:**
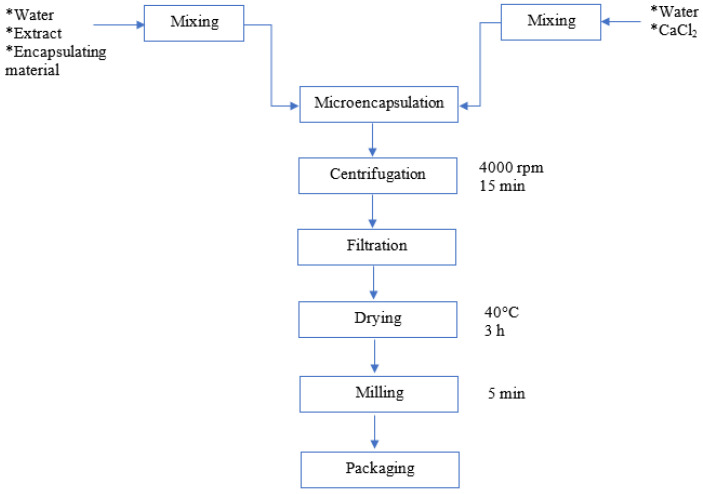
Flow chart for microencapsulation production.

**Figure 2 foods-12-02943-f002:**
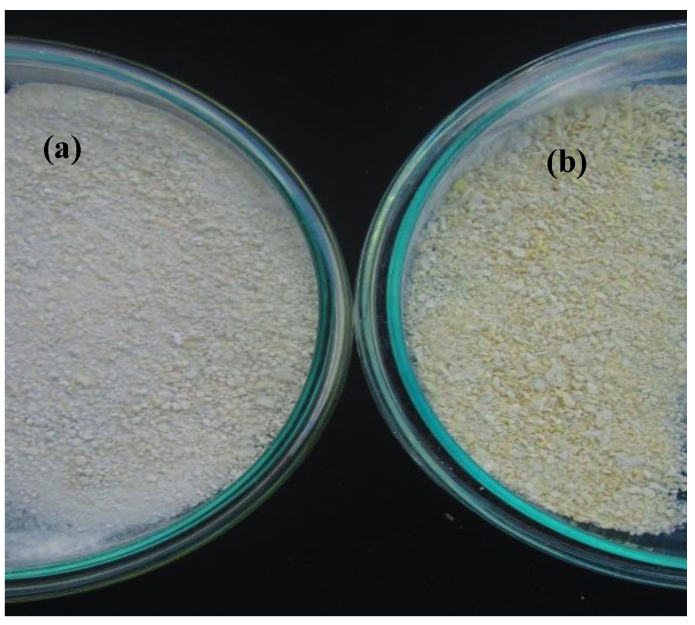
(**a**) Color of the microencapsulates without extract; (**b**) color of the microencapsulates with the extract.

**Figure 3 foods-12-02943-f003:**
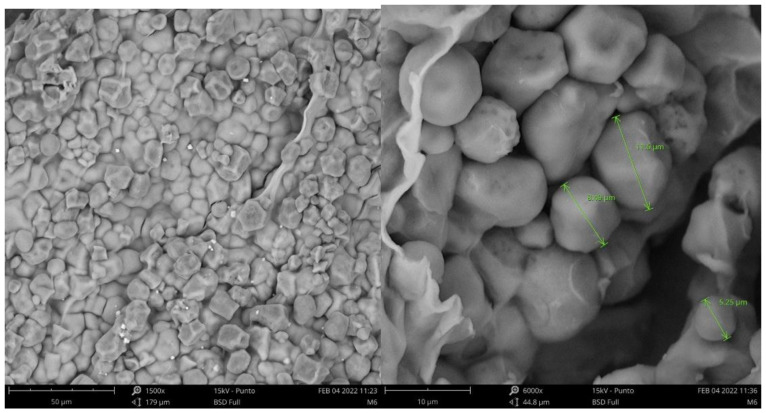
Scanning electron microscopy (SEM) images of particles obtained by extrusion drying.

**Figure 4 foods-12-02943-f004:**
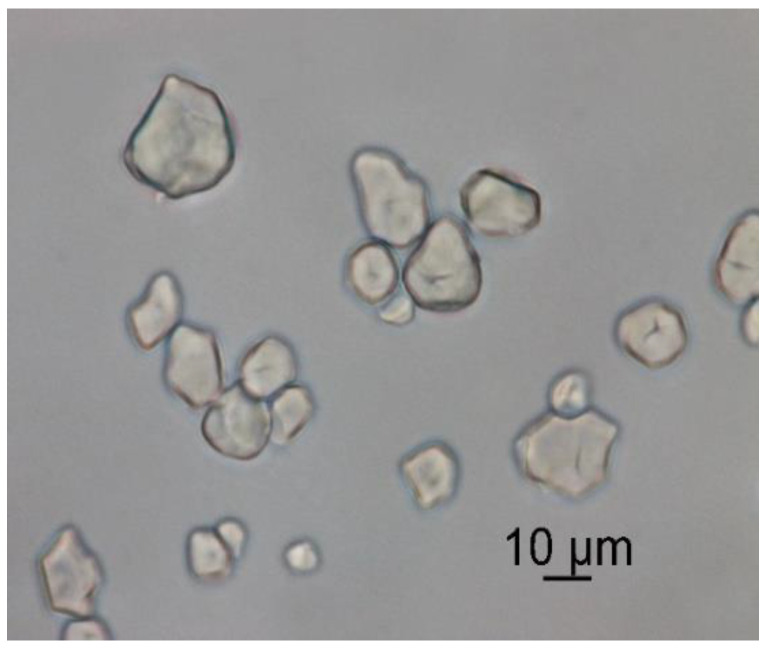
Image of the microencapsulates under 800×.

**Figure 5 foods-12-02943-f005:**
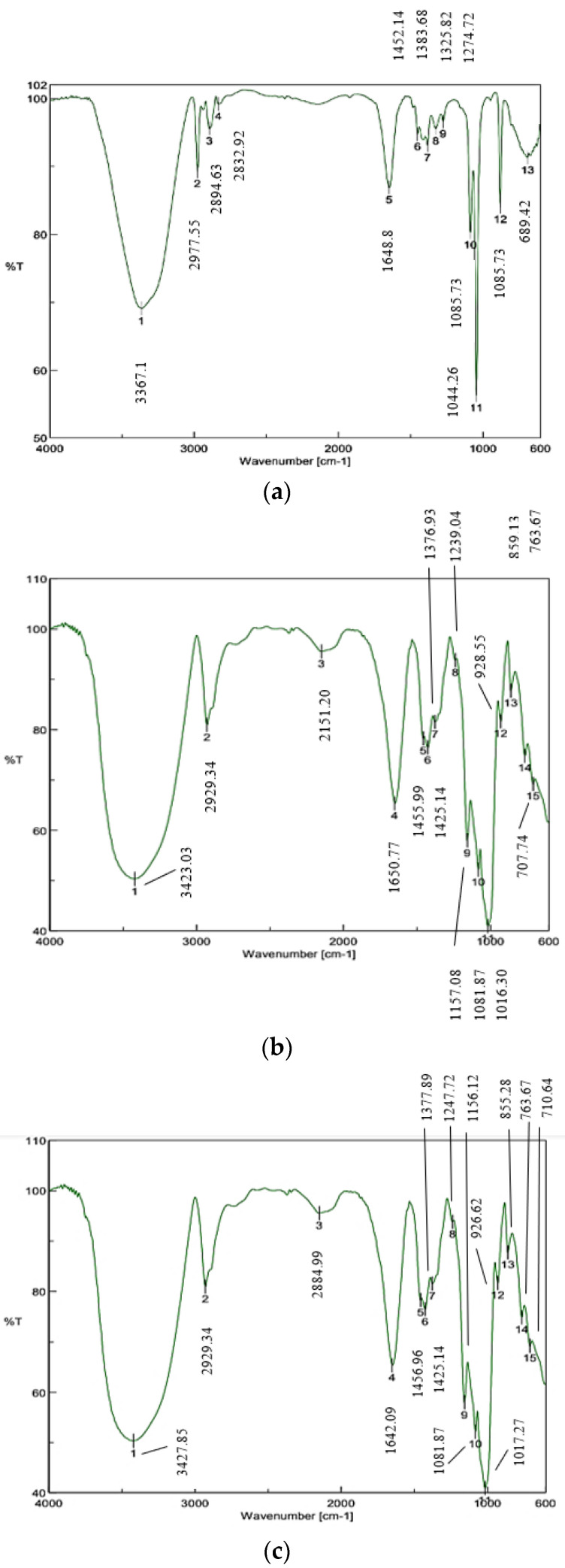
FTIR spectra of (**a**) extract, (**b**) wall materials, and (**c**) encapsulate.

**Figure 6 foods-12-02943-f006:**
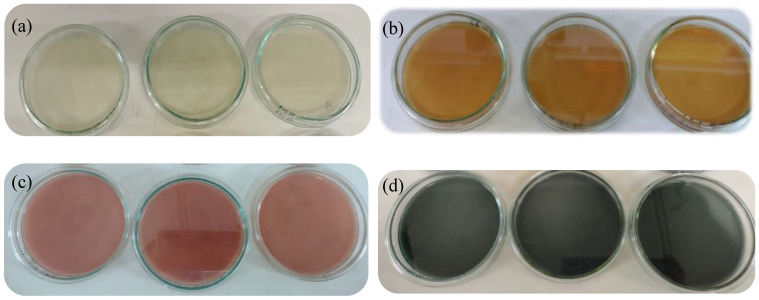
Microbiological results after 24 h of inoculation: (**a**) potato dextrose agar (PDA), (**b**) plate count agar (PCA), (**c**) violet red bile agar, and (**d**) MRS agar.

**Table 1 foods-12-02943-t001:** Physicochemical properties of the solvent-free extract.

Parameter		Results ^1^
Moisture content (w.b%)		95.90 ± 0.05
pH		3.85 ± 0.02
Color	L *	24.65 ± 0.13
	a *	1.75 ± 0.03
	b *	29.06 ± 0.14
Soluble solids	°Brix	10.13 ± 0.06
Total phenolic content	Folin–Ciocalteau (mg GAE/g d.m)	882.71 ± 2.07
Antioxidant activity	DPPH (% inhibition)	63.53 ± 1.10
	ABTS (% inhibition)	80.06 ± 2.18

^1^ Averages ± standard deviation (*n* = 3 replicates).

**Table 2 foods-12-02943-t002:** Results of the bioactive compounds present in the microencapsulates.

Compound	t_r_ (min)	MQL (mg/kg)	Concentration (mg/kg)
Theobromine	2.6	0.1	<0.1 *
Theophylline	2.8	0.1	<0.1 *
Epigallocatechin (EGC)	2.9	0.1	<0.1 *
Catechin ©	3.0	0.1	<0.1 *
Epicatechin (EC)	3.2	0.1	<0.1 *
p-Hydroxybenzoic acid	3.2	0.1	<0.1 *
Caffeine	3.1	0.1	<0.1 *
Caffeic acid	3.2	0.1	5.1 ± 0.1
Vanillic acid	3.2	0.1	5.7 ± 0.2
Rutin	3.2	0.1	0.1 ± 0.01
p-coumaric acid	3.6	0.1	2.6 ± 0.1
Epicatechin gallate (ECG)	3.6	0.1	<0.1 *
Ferulic acid	3.7	0.1	3.9 ± 0.2
Quercetin	4.5	0.1	0.5 ± 0.01
Rosmarinic acid	4.0	2.0	10.0 ± 0.2
Cyanidin	3.8	0.1	<0.1 *
Luteolin	4.4	0.1	<0.1 *
Kaempferol	4.9	0.1	<0.1 *
Trans-cinnamic acid	3.7	0.4	<0.1 *
Naringenin	4.8	0.1	<0.1 *
Pelargonidin	3.6	0.1	<0.1 *
Apigenin	4.8	0.1	<0.1 *
Pinocembrin	5.8	0.1	<0.1 *
Carnosic acid	7.3	0.4	<0.1 *
Ursolic acid	8.6	0.1	<0.1 *
Cyanidin 3-rutinoside	3.0	0.1	<0.1 *
Pelargonidin 3-glucoside	3.1	0.1	<0.1 *
Quercetin 3-glucoside	3.6	0.1	<0.1 *
Kaempferol 3-glucoside	3.8	0.1	<0.1 *
Rutin	3.5	0.1	<0.1 *
Gallic acid	1.8	0.1	1.7 ± 0.1

* Detected below the limit of quantification and above the limit of detection. MQL: Minimum quantification level. All values are expressed as means ± standard deviation.

**Table 3 foods-12-02943-t003:** Microbiological results of encapsulates, taking as reference the requirements for unmodified starch (NTC-926/2016).

Microorganism	Permissible Range	Results
Mesophilic aerobic bacterial count, CFU/g	10,000–50,000	Absence
Escherichia coli count, in CFU/g	<10	Absence
Fungi and yeast count, CFU/g	50–500	Absence

## Data Availability

The data used to support the findings of this study can be made available by the corresponding author upon request.

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
