# Peer review of "Encapsulation of Pineapple Peel Extracts by Ionotropic Gelation Using Corn Starch, Weissella confusa Exopolysaccharide, and Sodium Alginate as Wall Materials"

_foods, 2023, doi:10.3390/foods12152943_

Round 1

Reviewer 1 Report

The work presents the microencapsulation by ionotropic gelation and not by extrusion. The work is very interesting and completely characterizes the microencapsulation products. I recommend some minor changes.  

Please provide more details about:

- What is the modification of corn starch?

- more information regarding the EPS, expected molecular weight for instance. 

- did solid fermentation caused the complete digestion of cellulose? 

- please consider changing the name extrusion to ionotropic gelation. 

Reviewer 2 Report

The authors indicated the title of the manuscript as “Encapsulation of pineapple peel extract by extrusion using corn starch, Weissella confuse exopolysaccharide, and sodium alginate as wall materials.” They explained the pineapple, extraction of bioactives from pineapple, the need for the encapsulation of bioactives, extrusion as an encapsulation method. After that they must give the literature background on the corn starch, the exopolysaccharide and sodium alginate, respectively, as in the order of the title.  Or the order of the mention of the wall materials should depend on the concentration that was used in the study. And then their mention in the title and the text accordingly.

The sign of degree must be corrected as (o)   (lines 86, 193, 198, 378, 437, 438, 564, 571, 573, 637, 740).

Is there any characterization values/parameters for the exopolysaccharides? I think in order to determine the effect of exopolysaccharides, there should be known some of its parameters, such as viscosity at some concentration, etc.

In the microencapsulation process (line 142): The authors indicated that “The microencapsulation of the extract was performed the following extrusion method used by Turhan et al (32) and Shaharuddin & Muhammed (33) with some modification.” The authors must give the exact process conditions for the extrusion encapsulation. In addition, the amounts of sodium alginate, corn starch and exopolysaccharides along with the extract must also be given.

In the process yield (line 168): When the authors calculated the yield percent of encapsulates, they gave the Wextract was the mass of the extract entering the extrusion process and Wencapsulates was the mass of the encapsulate after drying. Then, Wencapsulates must contain the pineapple extract and the wall materials, however Wextract does not contain the wall materials, doesn’t it? Please clarify the calculation methods and the parameters the you used.

In the line 174: I think “y” should be corrected as “and”

 The section 2 materials and methods (line 92): The authors gave the materials where they were used in which methods, I think that it is confusing, because in the methods parts, the methods must be given clearly. Therefore, the all materials that were used in the study must be given in the 2.1. materials part. And then 2.2. is for the  “Methods”.

Line 99: Why does the extract inoculated with Rhizopus oryzae? I understand it was for fermentation but the authors should give the reason for it.

Line 344: “…the two wall materials used in greater proportion: corn starch and sodium alginate.” What are the amounts of these two wall materials?

 Line 382-385: The stability of encapsulated materials should be compared to the encapsulated systems which have the same wall materials. In addition, no imperfection is a good indicator for stability, however stability against other ambient conditions are also be taken into account. Furthermore (lines 391-394), the authors reported from the literature that high hygroscopicity leads to constant change in in the morphology. Then, how is this a stabile encapsulation? I think the authors must find another supportive study from the literature.

Line 439: The water activity values was given as the mean ± std dev. Both the mean value and the std dev. value must have the same decimal precision.

Line 440: must be corrected as “aw < 0.60”

Lines 549-551: The authors reported that “TPC was preserved in a higher proportion (33 times more) and 549 DPPH (2.4 times more), reflecting the advantages of the combined process of solid-state fermentation with ultrasound.” However they did not give any possible reason/mechanism for it. How come the total phenolic content and antioxidant activity increased after encapsulation? If the fermentation increased it, then the authors could measure the TPC and antioxidant activity after fermentation. Maybe encapsulation decreases them after fermentation. So, how come the authors conclude that encapsulation increases the TPC and DPPH?

Results-Encapsulation efficiency (line 578): The authors explained the high encapsulation efficiency based on the wall materials. However, they did not give the amount of the wall materials it is not clear that which one’s effect was the dominant. The authors compared the encapsulation efficiency studies from the literature with the ratios of wall materials.

In table 2: What does MQL stands for?

Based on the title, the conclusion would be evaluated in terms of the effects of wall materials. The authors gave the objectives in the beginning; one of them to show the effects of wall materials. It is not clear that the effects of wall materials are mainly due to their combination (starch+alginate+exopolysaccharide) or to their ratios or both.

The similarity report obtained from turnitin revealed that the similarity is 32 % without bibliographia. This must be reduced to below 20%.
